

# Intrasaccadic perception triggers pupillary constriction

Sebastiaan Mathôt, Jean-Baptiste Melmi and Eric Castet

Laboratoire de Psychologie Cognitive, UMR 7290, Aix-Marseille University/CNRS, Marseille, France

## ABSTRACT

It is commonly believed that vision is impaired during saccadic eye movements. However, here we report that some visual stimuli are clearly visible during saccades, and trigger a constriction of the eye's pupil. Participants viewed sinusoid gratings that changed polarity 150 times per second (every 6.67 ms). At this rate of flicker, the gratings were perceived as homogeneous surfaces while participants fixated. However, the flickering gratings contained ambiguous motion: rightward and leftward motion for vertical gratings; upward and downward motion for horizontal gratings. When participants made a saccade perpendicular to the gratings' orientation (e.g., a leftward saccade for a vertical grating), the eye's peak velocity matched the gratings' motion. As a result, the retinal image was approximately stable for a brief moment during the saccade, and this gave rise to an intrasaccadic percept: A normally invisible stimulus became visible when eye velocity was maximal. Our results confirm and extend previous studies by demonstrating intrasaccadic perception using a reflexive measure (pupillometry) that does not rely on subjective report. Our results further show that intrasaccadic perception affects all stages of visual processing, from the pupillary response to visual awareness.

## INTRASACCADIC PERCEPTION TRIGGERS PUPILLARY CONSTRICTION

When you look at yourself in the mirror, it seems that you make no eye movements. And that is odd, because everyone else does.

The inability to see your own saccadic eye movements is a simple yet powerful demonstration of saccadic suppression: You generally do not perceive events that occur while your eyes are moving, including (tautologically) your own saccades. Phrased differently, the retinal-image motion that results from saccades is suppressed, at least in the sense that it is not consciously perceived. Saccadic suppression is sometimes considered a mechanism that prevents you from seeing the world move when your eyes move (e.g., *Ross, Burr & Morrone, 1996*). However, this functional interpretation is controversial (e.g., *Castet, 2010*; *O'Regan & Noë, 2001*), and here we use saccadic suppression as an umbrella term for various findings that show that vision is impaired around the time of saccades.

Corresponding author
Sebastiaan Mathôt,
s.mathot@cogsci.nl

Saccadic suppression affects most if not all aspects of visual processing. Psychophysical studies have shown that visual sensitivity is reduced during a ±100 ms window around saccades (*Diamond, Ross & Morrone, 2000*; *Volkmann et al., 1978*; *Zuber & Stark, 1966*), especially for non-isoluminant stimuli with a low spatial frequency (*Burr, Morrone & Ross, 1994*; *Knöll et al., 2011*; but see *Braun, Schuetz & Gegenfurtner, 2015*). Neurophysiological studies have shown that stimuli presented during that same window elicit reduced neural activity in many visual brain areas (*Bremmer et al., 2009*; *Thiele et al., 2002*). Saccadic suppression even affects the pupillary light response: The pupillary constriction that is triggered by a flash of light is reduced when the flash occurs around the time of a saccade (*Lorber, Zuber & Stark, 1965*; *Zuber, Stark & Lorber, 1966*).

However, saccadic suppression is not complete; sometimes you can see while the eyes move, a phenomenon that we will refer to as *intrasaccadic perception*. Intrasaccadic perception can even be very conspicuous, as was elegantly shown by *Campbell & Wurtz (1978)*. In their experiment, participants sat in the dark and made a saccade. Crucially, if the experimental room was illuminated during the saccade, participants perceived a smeared-out image of the room—an intrasaccadic percept. But this only happened when the illumination period fell entirely within the saccade. If the room was also briefly illuminated before or after the saccade, smearing was no longer perceived; instead, participants perceived a static image of the room. In other words, intrasaccadic perception occurred only when there was no pre- or postsaccadic visual stimulation; otherwise, the intrasaccadic percept was not consciously perceived, presumably because it was masked by the pre- and postsaccadic percept (reviewed in *Castet, 2010*). Intrasaccadic perception has also been demonstrated with stimuli that move too rapidly to be seen with static eyes (e.g., a vertical grating that moves across a display with a speed of 400°/s). Saccades can make such stimuli visible by reducing retinal-image motion: When you make a saccade in the direction of the stimulus' movement (i.e., perpendicular to the grating's orientation), the velocity of the saccade briefly reduces, or even cancels, the retinal speed of the stimulus, which is consequently perceived as a static or moving percept, depending on the stimulus and the eye movement's velocity (*Castet & Masson, 2000*; *Deubel & Elsner, 1986*; *García-Pérez & Peli, 2001*). Phrased differently, a saccade can make a normally invisible stimulus visible by briefly reducing, or even canceling, its retinal speed. This is a striking phenomenon that we will use in the present study as well. (We also provide demonstration software, as described under Methods.)

To date, intrasaccadic perception has only been demonstrated through subjective report. In a typical experiment, participants first make a saccade, and then indicate whether or not they perceived something (a flash or a movement) during the saccade. Here we use pupillometry to provide a direct demonstration of intrasaccadic perception that does not rely on subjective report. To this end, we used a paradigm similar to that used by *Castet & Masson (2000)*, and measured pupillary responses, which are increasingly recognized as a powerful tool for vision science (for recent reviews, see *Binda & Murray, 2014*; *Mathôt & Van der Stigchel, in press*). Under normal conditions, centrally presented visual stimuli trigger a pupillary constriction, even when there is no change in overall

luminance (e.g., *Gamlin et al., 1998*; *Slooter & Van Norren, 1980*; *Ukai, 1985*). Therefore, we predicted that an intrassacadic stimulus, which in our view is not fundamentally different from any other stimulus, would also trigger a pupillary constriction (or rather a strengthening of the pupillary constriction that is generally observed after a saccade; *Mathôt et al., 2015*; *Zuber, Stark & Lorber, 1966*).

If we would find a pupillary constriction to intrasaccadic perception, this would be important for two main reasons. Firstly, our study avoids expectation effects that were not addressed previously: The knowledge that you may see something during a saccade may by itself increase the saliency of the intrasaccadic percept, and, in the extreme case, even induce a 'placebo percept.' Therefore, we tested only naïve observers, and we did not inform them beforehand that they might perceive 'something odd' during a saccade. Secondly, a pupillary constriction to intrasaccadic perception would show that intrasaccadic perception is similar to regular perception, not only in eliciting a conscious percept, but also in triggering a reflexive pupillary response.

## METHODS

### Materials and availability

Participant data, the experimental script, and analysis scripts are available from https://github.com/smathot/materials_for_P0018. On-line materials include easy-to-use demonstration software that allows anyone with a sufficiently fast display ($>= 150$ Hz) to experience intrasaccadic perception.

### Participants

Ten naïve observers (age range: 18–21 y; 3 men) participated in the experiment. Participants reported normal uncorrected vision. Participants provided written informed consent prior to the experiment, and received € 20 for their participation. The experiment was conducted with approval of the *Comité d'éthique de l'Université d'Aix-Marseille* (Ref.: 2014-12-03-09).

### Software and apparatus

Eye position and pupil size were recorded monocularly with an EyeLink 1,000 (SR Research, Mississauga, Ontario, Canada), a video-based eye tracker (1,000 Hz; ellipse mode). Stimuli were presented on a gamma-calibrated 21″ ViewSonic p227f CRT monitor (1,024 × 768, 150 Hz). Testing took take place in a dimly lit room. The experiment was implemented with OpenSesame (*Mathôt, Schreij & Theeuwes, 2012*) using the PsychoPy back-end (*Peirce, 2007*) for display control, and PyGaze (*Dalmaijer, Mathôt & Van der Stigchel, 2014*) for eye tracking.

### General stimuli and procedure

At the beginning of each session, a nine-point eye-tracker calibration was performed. Before each trial, a small dot was presented 8.5° left of, right of, above, or below the display center. When a stable fixation was detected, a one-point recalibration (drift correction) was performed. Next, the trial started with the presentation of a green fixation dot

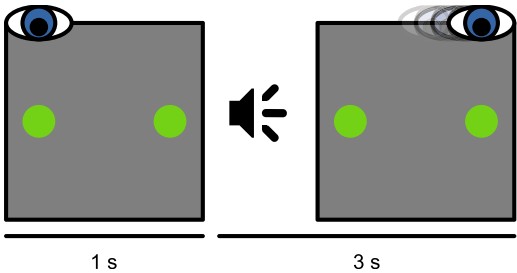

1 s          3 s

**Figure 1 Schematic experimental paradigm.** Participants initially fixated on a dot presented at the left, right, top, or bottom of the screen. A saccade target was presented at the location opposite from the fixation dot. An auditory cue instructed participants to make a saccadic eye movement to the saccade target.

(36.8 cd/m$^2$) at the same location as the drift-correction stimulus. At the same time, another green dot (the saccade target) was presented at the location opposite from the fixation dot (Fig. 1). After 1 s, an auditory cue (a 100 ms, 440 Hz sine wave) instructed the participant to make a saccade from the initial fixation dot to the saccade target. The trial ended 3 s after the onset of the auditory cue. Participants were instructed not to move their eyes before the cue, and to fixate on the saccade target until the end of the trial. In total, the experiment lasted about 2.5 h. For some participants, data collection was spread across two days, in which case peak saccade velocities (as described below) were determined again on the second day.

### Part 1: Peak saccade velocity

In the first part of the experiment, we determined the median peak velocity of saccades for each direction (left, right, down, up) and participant. The background was static and uniformly gray (51.1 cd/m$^2$). Gaze position was sampled on every frame (i.e., every 6.67 ms). Blinks were treated as missing data. On every trial, the maximum of the peaks of the horizontal or vertical velocity profiles for the entire trial was taken as the peak saccade velocity; that is, peak velocity was the highest speed of the eyes at any moment and in any direction. When peak saccade velocity was unrealistic (above 664°/s or below 220°/s, based on a peak velocity of around 400°/s for 17° saccades, see *Baloh et al., 1975*), the trial was discarded and repeated at a random moment during the remainder of the session. For every saccade direction, peak saccade velocity was based on the median of 40 trials.

### Part 2: Intrasaccadic perception

In the second part of the experiment, we investigated the effect of intrasaccadic perception on pupil size. The background was a 22.6° × 22.6° full-contrast sinusoid (5.2 cd/m$^2$ to 95.1 cd/m$^2$) grating that reversed in polarity on every frame (Fig. 2). During fixation, the background appeared a homogeneous gray surface, because the frame rate (150 Hz) exceeded the flicker fusion threshold (see Fig. 3A). Furthermore, the display appeared uniformly gray, because the monitor was gamma calibrated: A pixel that alternated between minimum (5.2 cd/m$^2$) and maximum brightness (95.1 cd/m$^2$) had the same mean (fused) luminance as a pixel that was constantly at 50% brightness (50.2 cd/m$^2$). The

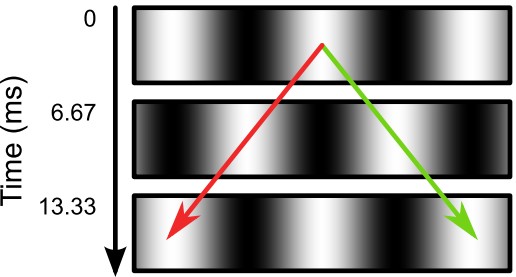

Perceived leftward motion
Perceived rightward motion

**Figure 2 Illustration of ambiguous motion.** During Part 2 of the experiment (see main text), the background consisted of a sinusoid grating that reversed polarity on every frame. The motion direction in this flickering display is ambiguous. The two opposite motion signals (leftward: red arrow; rightward: green arrow) result from matching the white and black bars across time. When viewed with static eyes, the flickering display appears as a homogeneous gray surface.

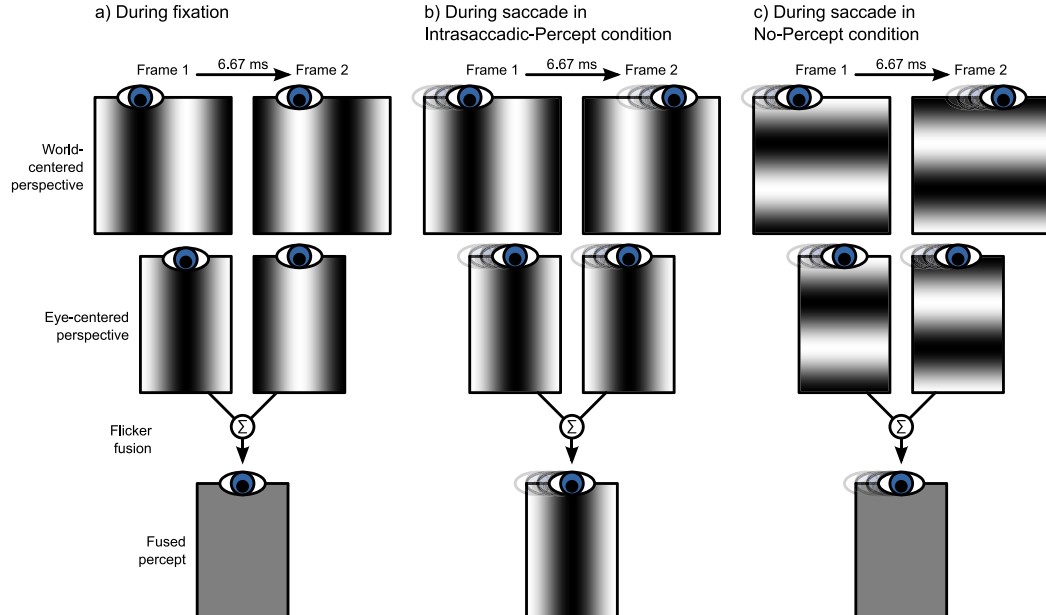

**Figure 3 Illustration of intrasaccadic perception.** (A) The background is a rapidly flickering grating that is perceived as uniformly gray during fixation. (B) When a saccade is made perpendicular to the grating (Intrasaccadic-Percept condition), the grating is briefly stabilized on the retina, which results in a flash-like intrasaccadic percept. (C) When a saccade is made parallel to the grating (No-Percept condition), the grating is not stabilized on the retina, and no intrasaccadic percept arises.

spatial frequencies of the gratings were set for each participant and saccade direction separately, based on the peak saccade velocities estimated during the first part of the experiment. (Due to a technical issue, for three participants the gratings were set to a default spatial frequency of 0.17 cycles/°. However, this default value was based on pilot testing, and was close to the ideal spatial frequency.) The spatial frequency was such that the maximum distance that the eyes traveled between two frames was equal to half a cycle.

The logic behind this is that the flickering background contains ambiguous motion: It can be perceived as moving leftward or rightward with a speed of half a cycle per frame (Fig. 2). Therefore, the peak velocity of the eyes matched the velocity of the motion. As a result, the background approximated a retinotopically stabilized grating when saccade velocity was maximal (Fig. 3B).

In the Intrasaccadic-Percept condition, the orientation of the grating was perpendicular to the direction of the saccade. Therefore, in this condition, participants should briefly perceive a static grating during the saccade, as described above (Fig. 3B). In the No-Percept condition, the orientation of the grating was parallel with the direction of the saccade. Therefore, in this condition, no static grating should be perceived, because the saccade did not stabilize the grating on the retina (Fig. 3C).

## Part 3: Subjective report

At the end of the experiment, participants provided a subjective report of how strongly they had perceived 'something odd' while making saccades. They provided separate ratings for each saccade direction on a 1–5 scale. Participants did not report the nature of the percept, i.e., whether they had perceived static or moving gratings.

# DATA PREPROCESSING AND STATISTICAL ANALYSES

## Preprocessing of pupil and eye-position signal

All signals were locked to the moment that the eyes crossed the vertical (for horizontal saccades) or horizontal (for vertical saccades) meridian (from now on: mid-saccade point). Mean pupil size from 105 to 95 ms before the mid-saccade point was takes as a baseline, and all pupil size measures are reported in area (i.e., not diameter) relative to this baseline (cf. *Mathôt et al., 2015*). We analyzed pupil size from 300 ms before until 1,200 ms after the mid-saccade point. Pupil size during blinks was reconstructed using cubic-spline interpolation (*Mathôt, 2013*). Pupil size was not smoothed. To obtain an acceptable noise level in eye-velocity profiles, eye position was smoothed using an 11 ms Hanning window.

## Trial-exclusion criteria

Trials were discarded when any of the following criteria was met: saccade latency was below 0 ms (i.e., anticipation) or above 2,000 ms; peak saccade velocity could not be determined or was unrealistically high ($>1,000°$/s; usually due to data loss); or the eyes deviated more than $3.3°$ from the fixation dot before the saccade (excluding a 200 ms around the mid-saccade point). The eyes deviated more than $3.3°$ from the saccade target after the saccade (again excluding a 200 ms around the mid-saccade point). No participants were excluded. In total, 4,001 trials (83.3%) remained for further analysis.

## Position artifacts in pupil size

In video-based eye trackers, changes in eye position cause artifactual changes in measured pupil size. That is, when the eyes move, the angle from which the camera records the eye changes, and this may lead the eye tracker to register changes in pupil size, even when pupil size remains constant. A common way to correct for position artifacts is to determine a

linear regression that predicts (baseline) pupil size from horizontal ($X$) and vertical ($Y$) eye position. The $X$ and $Y$ slopes can then be used to 'regress out' eye position from pupil size (e.g., *Brisson et al., 2013*).

In our data, position artifacts are visible as rapid changes in pupil size during the saccade (Figs. 5A, 5D, 5G and 5J); these changes are too rapid to be real. But when we tried to eliminate position artifacts through linear regression, we noticed the following: When fixating the top of the screen, the pupil was larger than when fixating the bottom of the screen (data not shown), which suggests that pupil size is *overestimated* when fixating the top of the screen. But when participants made a top-to-bottom saccade, pupil size increased even further (Fig. 5J), which suggests that pupil size is *underestimated* when fixating the top of the screen. In other words, the direction of the position artifact seems to depend on whether you look at baseline pupil size, or at the rapid change that occurs during the saccade. This implies that there are two distinct effects of eye position on pupil size as measured by the eye tracker: The first is a real effect, so that (in our case) the pupil slightly dilates when fixating the top of the screen, perhaps due to effort or the eye's musculature; the second is an artifact, so that (in our case) the size of the pupil is underestimated when fixating the top of the screen. We are currently investigating this issue further, but for now we note that: not all effects of eye position on pupil size are artifactual (as generally assumed); and any corrective technique that assumes this is problematic.

Therefore, we decided not to correct our pupil-size measurements for position artifacts, and absolute changes in pupil size that occur while the eyes are moving (as shown in Figs. 5A, 5D, 5G and 5J) should be interpreted with caution.

## Statistical analyses

Unless otherwise specified, we used linear mixed-effects models (LME) with by-participant random intercept and by-participant random slopes for all predictors. Identical analyses were conducted separately for each 1 ms sample. We did not estimate $p$-values, but considered effects reliable if they correspond to $t > 2$ for at least 200 consecutive 1 ms samples (cf. *Mathôt et al., 2014*). However, we emphasize effect sizes and overall patterns.

# RESULTS

## Intrasaccadic perception induced pupillary constriction

The main result is shown in Fig. 4A, in which pupil size is plotted over time as a function of condition (Intrasaccadic Percept vs No Percept). The pupil initially dilated in both conditions, until about 200 ms after the mid-saccade point, reflecting motor preparation (*Jainta et al., 2011*; *Wang, Brien & Munoz, 2015*). Next, the pupil constricted in both conditions from about 220 ms after the mid-saccade point. Such a constriction is typically observed after saccades (*Mathôt et al., 2015*; *Zuber, Stark & Lorber, 1966*). Crucially, as predicted, this constriction was more pronounced in the Intrasaccadic-Percept Condition than in the No-Percept Condition. Based on an LME with pupil size as dependent measure, and Condition (reference: No Percept), Saccade Direction (reference: Horizontal),

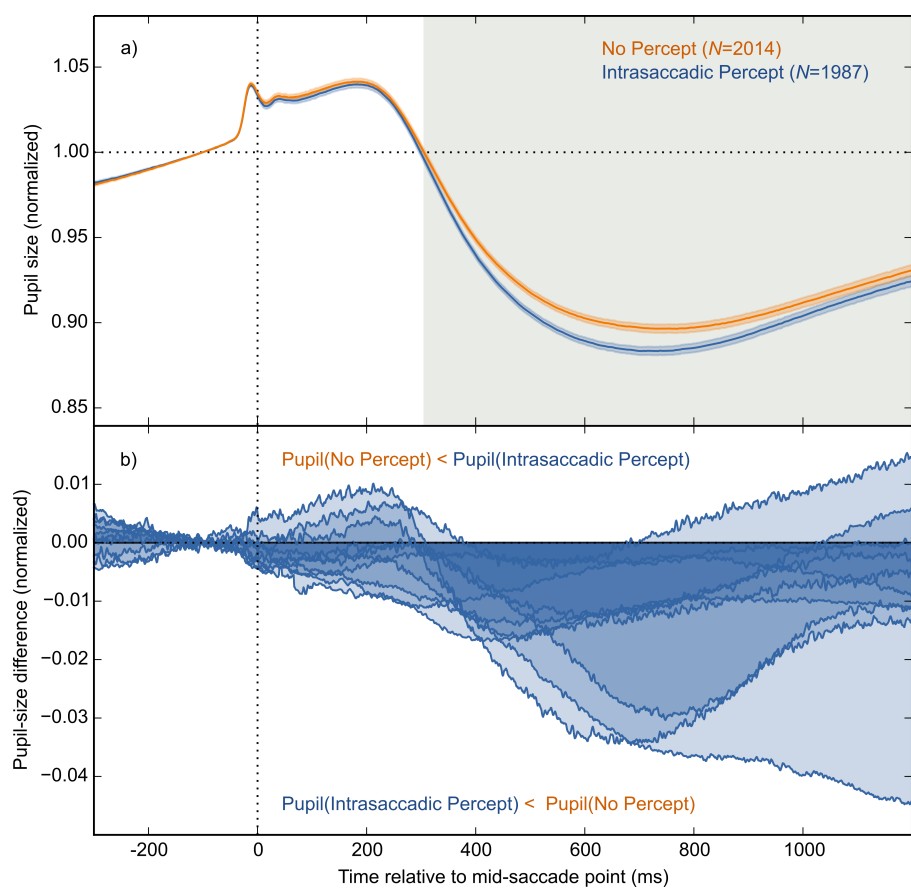

**Figure 4 Main results.** (A) Mean normalized pupil size over time, locked to the mid-saccade point. There was a more pronounced constriction in the Intrasaccadic Percept Condition than in the No-Percept Condition. Error bands indicate the standard error. Gray shading indicates a reliable effect of Condition. (B) The pupil-size difference between the Intrasaccadic Percept and No-Percept conditions for each individual participant.

and their interaction as fixed effects, the effect of condition was reliable from 306 ms after the mid-saccade point until the end of the analysis period. As can be seen in the indivual-participant data (Fig. 4B), this effect was highly consistent across participants.

## Differences between saccade directions

Based on the LME described above, there was a reliable interaction between Condition and Saccade Direction from 426 ms until the end of the analysis period. This interaction indicated that the Condition effect was driven mainly by horizontal saccades. We confirmed this by analyzing each saccade direction separately, based on four separate LMEs with Condition as fixed effect and pupil size as dependent measure (Fig. 5). There was a strong and reliable Condition effect for leftward and rightward saccades (Figs. 5A and 5D). There was also a reliable Condition effect for downward saccades (Fig. 5G), but it was much weaker. For upward saccades, there was no reliable Condition effect (see Fig. 5J). In addition, there was considerable sample-to-sample variability in the model estimates for

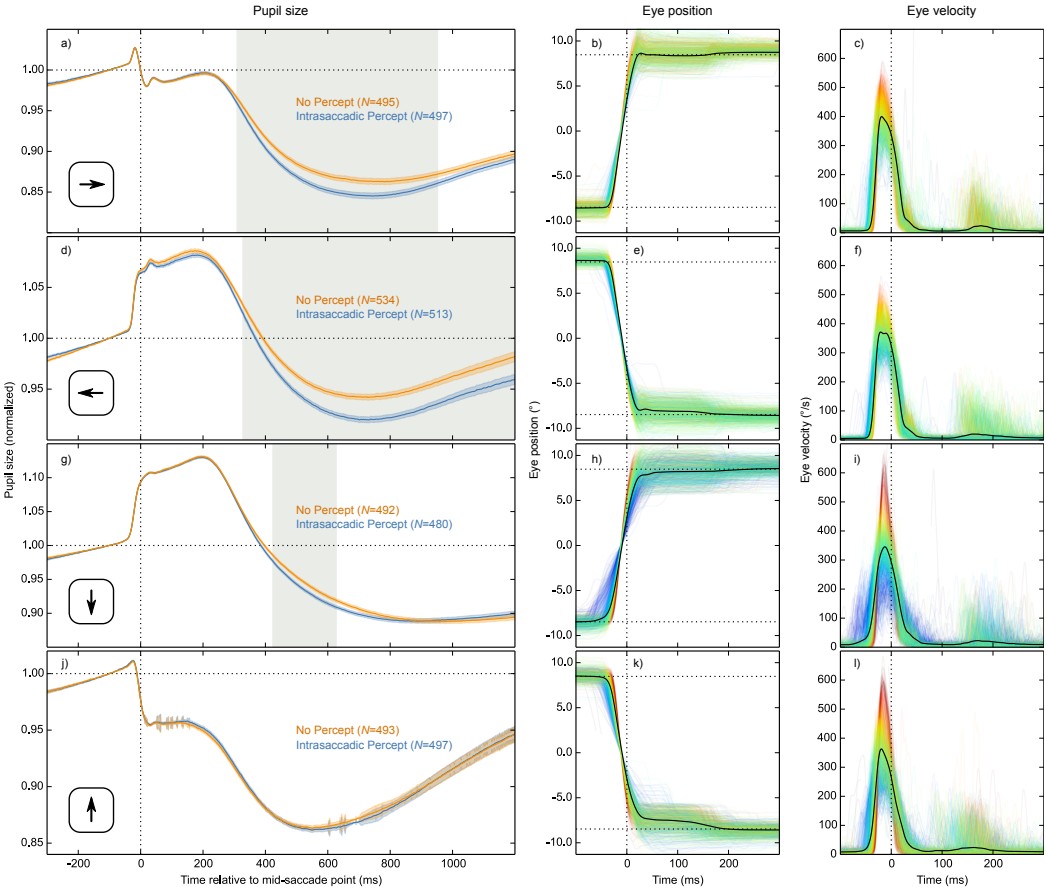

**Figure 5 Results split by saccade direction.** Pupil-size, eye-position, and eye-velocity traces for each of the four saccade directions. The large differences in pupil size just before and after saccades between the four saccade directions are due to position artifacts in pupil-size measurements (see main text). (A, D, G, J) Error bands indicate the standard error. Gray shading indicates a reliable effect of condition. See main text for a description of the statistical models. (B, E) Horizontal eye position over time. (C, F) Horizontal eye velocity over time. (H, K) Vertical eye position over time. (I, L) Vertical eye velocity over time. (B, C, E, F, H, I, K, L) Individual trials are color coded to indicate peak saccade velocity (red: high velocity; blue: low velocity).

upwards saccades (as shown by the jittery error bands in Fig. 5J), suggesting that this subset of data was especially noisy.

Although there are known differences between saccades in different directions (*Collewijn, Erkelens & Steinman, 1988*; *Van der Stigchel & Theeuwes, 2008*), we had not expected saccade direction to interact with the effect of intrasaccadic perception on pupil size. Therefore, we conducted several post-hoc analyses to better understand these differences.

First, peak velocities are more variable for vertical than horizontal saccades (compare Figs. 5C, 5F, 5I and 5L). This was the case for all participants (two-sided paired-samples $t$-test using per-participant standard deviation of peak velocity as dependent measure: $t(9) = 6.42$, $p = .0001$). This variability is important, because the intrasaccadic percept was optimized for a peak velocity of around 400°/s (depending slightly on the participant

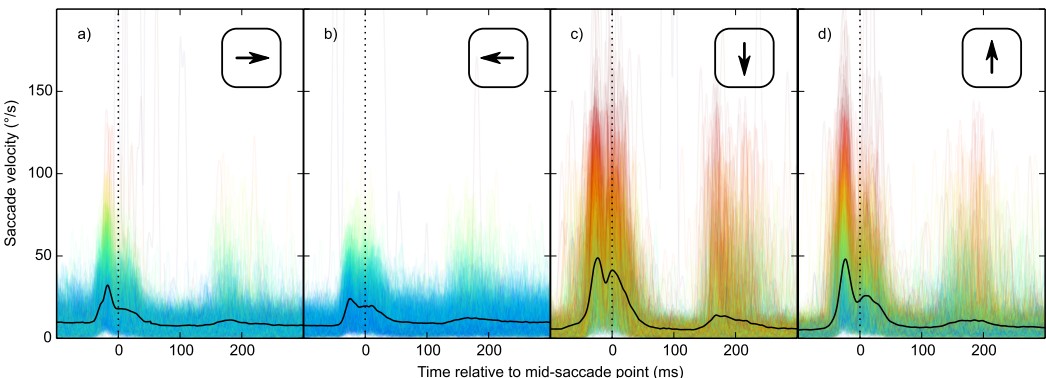

**Figure 6 Orthogonal eye velocity.** Eye velocity perpendicular to the saccade direction (orthogonal velocity). (A, B) Vertical eye velocity during horizontal saccades. (C, D) Horizontal velocity during vertical saccades. Individual trials are color coded to indicate peak orthogonal velocity (red: high velocity; blue: low velocity).

and saccade direction). Therefore, the intrasaccadic percept may have been less salient for vertical than horizontal saccades. To control for this, we determined the difference between the actual and the optimal peak velocity (from now on: peak-velocity error) for each trial. Next, we selected trials on which peak-velocity error was less than the median peak-velocity error (separately for each participant and saccade direction).

Second, vertical saccades are more curved than horizontal saccades (*Van der Stigchel & Theeuwes, 2008*). Therefore, the velocity component that is perpendicular to the saccade direction (i.e., horizontal velocity during vertical saccades, and vertical velocity during horizontal saccades; from now on: orthogonal velocity) was higher for vertical than horizontal saccades (Fig. 6). This was the case for all participants (two-sided paired-samples $t$-test using per-participant mean peak orthogonal velocity as dependent measure: $t(9) = 10.25$, $p < .0001$). Peak orthogonal velocity during vertical saccades often approached 150°/s. In the No Percept condition, this would have led to partial retinal stabilization of the grating, which is sufficient to trigger an intrasaccadic percept (*García-Pérez & Peli, 2001*). In other words, intrasaccadic perception may have triggered a pupillary constriction in both conditions, thus reducing the Condition effect. To control for this, we selected trials on which peak orthogonal velocity was less than the median peak orthogonal velocity (seperately for each participant and saccade direction).

Next, using vertical saccades from this subset of data, selected as described above, we performed the same analysis as before. Based on an LME with Condition as fixed effect and pupil size as dependent measure, we now also observed a reliable Condition effect for vertical saccades (Fig. 7). Therefore, it is likely that the Condition effect for vertical saccades was reduced by variability in peak velocity, and pronounced curvature.

### Subjective ratings

Figure 8 shows the participants' subjective ratings of how strongly they had perceived 'something odd' during saccades, separately for each saccade direction. These ratings suggest that all participants experienced an intrasaccadic percept, despite the fact that all

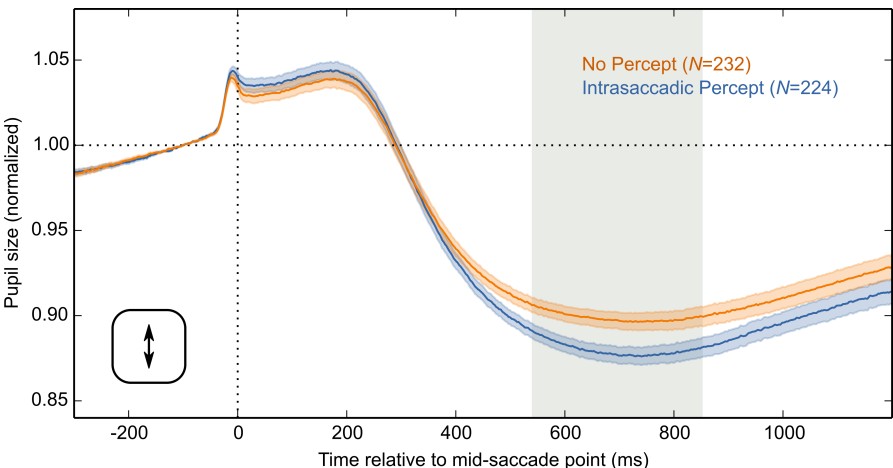

**Figure 7 Results for an optimal subset of vertical saccades.** Analysis of a subset of vertical-saccade trials on which orthogonal velocity was low and peak saccade velocity was close to optimal. Pupil size is plotted over time, locked to the mid-saccade point. There was a more pronounced constriction in the Intrasaccadic Percept Condition than in the No-Percept Condition. Error bands indicate the standard error. Gray shading indicates a reliable effect of Condition.

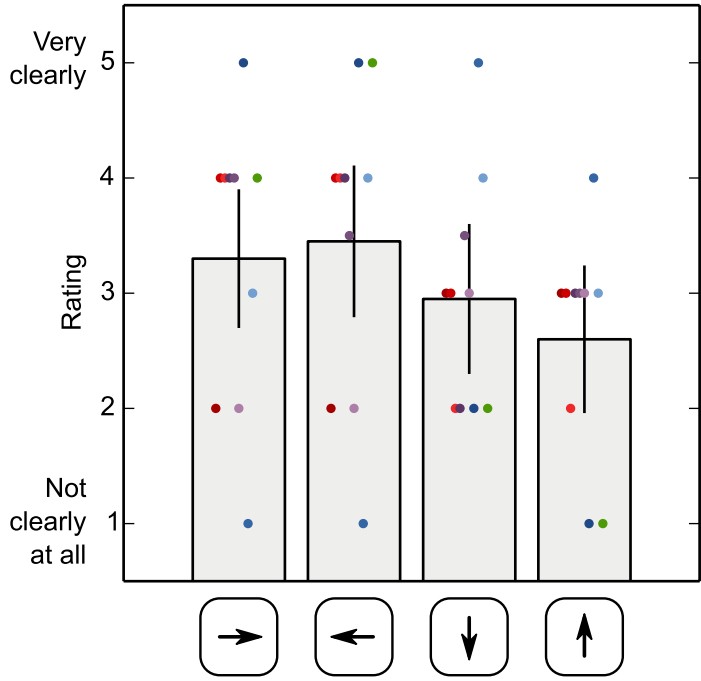

**Figure 8 Subjective ratings of intrasaccadic perception.** Participants' ratings of how strongly they had perceived 'something odd' during saccades, separately for each saccade direction. Error bars indicate 95% within-subject confidence intervals (*Cousineau, 2005*). Dots correspond to single ratings, color coded by participant.

participants were naïve and intrasaccadic perception was not mentioned until the end of the experiment. There is a tendency for the intrasaccadic percept to be rated more salient for horizontal than vertical saccades, but this tendency is weak and unreliable ($F(3, 27) = 0.918$, $p = .446$; based on a Repeated Measures Analysis of Variance with saccade direction as independent variable and rating as dependent measure).

## DISCUSSION

Here we report that intrasaccadic perception (seeing while the eyes move) triggers a constriction of the eye's pupil. We presented participants with sinusoid gratings that changed in polarity 150 times per second, or every 6.67 ms. At this speed, successive frames cannot be distinguished, and the flickering gratings were therefore perceived as homogeneous gray surfaces (Fig. 3A). However, the flickering gratings contained two opposite motion signals (Fig. 2). In one condition, there was a brief moment during each saccade at which the eye's velocity matched the direction and speed of one of the grating's motion signals (Fig. 3B). This resulted in a stable retinal image, in turn resulting in a brief-but-clear intrasaccadic percept. Relative to a control condition in which the saccade direction did not match the motion signal, this intrasaccadic percept triggered a slight pupillary constriction (Fig. 4).

To our knowledge, our study provides the first direct evidence for intrasaccadic perception that does not rely on subjective report. We did collect subjective ratings, which confirmed that participants saw an intrasaccadic percept (Fig. 8), but these ratings were collected after the experiment. Furthermore, our results show that intrasaccadic perception affects all stages of visual processing: from the reflexive pupillary response, as we report here, to conscious perception, as reported previously (*Campbell & Wurtz, 1978*; *Castet & Masson, 2000*; *Castet, Jeanjean & Masson, 2002*; *Deubel & Elsner, 1986*; *García-Pérez & Peli, 2001*). The picture that emerges is that there is nothing special about intrasaccadic perception. Visual perception may not be fundamentally different when the eyes move, compared to when the eyes do not move.

So far, we have focused on the pupillary constriction that is triggered by intrasaccadic perception (i.e., the difference between lines in Fig. 4). But saccades are usually followed by a pronounced constriction, also without intrasaccadic perception (i.e., the overall negative deflection in Fig. 4). Zuber and colleagues (*1966*; see also *Mathôt et al., 2015*) already noticed this, but attributed it to changes in accommodation that might accompany gaze shifts (i.e., the pupil near reflex). However, based on recent(ish) advances in pupillometry, we suggest that post-saccadic pupillary constriction may directly reflect intrasaccadic perception—both when consciously perceived, and when not consciously perceived.

As first documented by *Van de Kraats, Smit & Slooter (1977)*, any kind of change in visual input triggers a pupillary constriction, even when overall brightness does not change. For example, when you look at a checkerboard that changes polarity (i.e., the dark squares become bright, and vice versa), your pupil briefly constricts, before returning to its normal size (*Slooter & Van Norren, 1980*; *Ukai, 1985*). The origin of this phenomenon is unclear. According to *Barbur, Keenleyside & Thomson* (*1987*; cited in *Sahraie & Barbur, 1997*), it

may result from changes in neural activity that occur when visual input changes. This may cause a temporary reduction of the inhibition of the pathway that drives the pupillary sphincter muscle, thus causing a brief constriction. Alternatively, as we have recently suggested, pupillary constriction to visual change may simply reflect the asymmetry of the pupillary light response (*Mathôt et al., 2013*): Dilation in response to darkness is slower than constriction in response to brightness. Because visual changes are generally a mix of local increases and decreases in brightness, the combined pupillary response may be an initial constriction (to local brightness increases) that disappears as dilation (to local brightness decreases) catches up. But, whatever the cause, it is clear that the pupil constricts in response to changes in visual input.

Our interpretation of the postsaccadic pupillary response (Fig. 4) is therefore as follows. We assume that pupillary constriction to visual change is largely reflexive, but can be modulated by top-down factors, such as visual awareness. When the eyes move, visual input changes, and this triggers a pupillary constriction, just like a checkerboard inversion does. Usually, these saccade-related visual changes are not consciously perceived; but when they are, visual awareness increases the pupillary constriction. In other words, the slight additional constriction that is triggered by intrasaccadic perception may reflect top-down enhancement of the pupillary constriction to visual change. Although this interpretation is speculative, it fits with recent findings that pupillary responses, even those that were traditionally seen as pure reflexes, are susceptible to top-down modulation (reviewed in *Mathôt & Van der Stigchel, in press*).

In summary, we have shown that intrasaccadic perception triggers a pupillary constriction, or rather strengthens the pupillary constriction that is generally observed after saccadic eye movements. To our knowledge, this is the first direct evidence for intrasaccadic perception that does not rely on subjective report: Regardless of whether you expect it or not, intrasaccadic perception is a salient event that affects all stages of visual processing, from the pupillary response to visual awareness.

### Funding

The research leading to these results has received funding from the People Programme (Marie Curie Actions) of the European Union's Seventh Framework Programme (FP7/2007-2013) under REA grant agreement no. 622738. The funders had no role in study design, data collection and analysis, decision to publish, or preparation of the manuscript.

### Grant Disclosures

The following grant information was disclosed by the authors:
The authors declare there are no competing interests..

### Competing Interests

We declare no competing interests.

## Author Contributions

- Sebastiaan Mathôt conceived and designed the experiments, analyzed the data, wrote the paper, prepared figures and/or tables, reviewed drafts of the paper.
- Jean-Baptiste Melmi conceived and designed the experiments, performed the experiments, wrote the paper, reviewed drafts of the paper.
- Eric Castet conceived and designed the experiments, wrote the paper, reviewed drafts of the paper.

## Human Ethics

The following information was supplied relating to ethical approvals (i.e., approving body and any reference numbers):

The experiment was conducted with approval of the Comité d'éthique de l'Université d'Aix-Marseille (Ref.: 2014-12-03-09).

## Data Deposition

The following information was supplied regarding the deposition of related data:

https://github.com/smathot/materials_for_P0018.

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
