# Peer review of "Intrasaccadic perception triggers pupillary constriction"

_PeerJ, doi:10.7717/peerj.1150_

## Round 0.1 · original submission · Major Revisions

Dear Authors,

Please consider the revisions required to the manuscript according to the comments of both peer reviewers.

Thanking you.

·

Basic reporting

The basic structure of the paper, and of the study it describes, conforms to the highest standards of scientific publications.

Experimental design

The question is clear, meaningful and addressed with a technically sound (and very interesting) methodology; the latter is described in sufficient detail.

Validity of the findings

Results are clear, analyses are statistically sound and they fully support the man conclusions (with one caveat on the interpretation, see below my general comments)

Comments for the author

A saccade may reveal the real nature of an apparently homogeneous grey field, which in fact is a large grating that flickers above the fusion threshold of the fixating eye. To a naive observer, it will appear that “something odd” has happened during the saccade. This study is an interesting example of how pupillometry may be used to document such phenomenon without relying on participants’ introspection, in an objective and quantitative way. It is reported that the pupil responds differently in trials where the conditions for an ‘intra-saccadic’ percept are set, compared to when nothing but the grey screen may be perceived. Specifically, there is a more pronounced constriction in ‘intrasaccadic perception’ conditions.

There are only a few suggestions that I would like to offer, regarding the presentation and interpretation of the findings.


1. intra-saccadic perception should not be construed as disconfirming the “active suppression” hypothesis. I strongly suggest that the authors revise the couple of sentences in the intro/discussion where they make this claim.
No paper (that I know of) proposing that vision during saccades is actively suppressed claims that such suppression is complete. First off, several papers document that active suppression only affects some stimuli, leaving visibility of others perfectly unaltered, e.g. high spatial frequencies (Burr Morrone Ross 1994) and chromatic modulations (e.g. Knoll, Binda, Morrone and Bremmer JoV 2011). Even for the stimuli that are maximally affected, like luminance modulations at low spatial frequency, suppression is never claimed to be total - only a reduction of sensitivity. Of course, the terms high/low spatial/temporal frequency refer to the retinal stimulus, not the stimulus presented on the screen. According to the active suppression idea, one would predict that stimuli generating the same retinal stimulation are less visible (elicit smaller pupil changes) during saccades than in fixation. Matching retinal stimulation can be achieved, for example, by using a rotating mirror to simulate the eye movement and reproduce the quasi-stabilized intra-saccadic gratings used here. Lacking such comparison, speaking of intrasaccadic enhancement seems misleading to me. It may be a useful description of the data, but then its relation to saccadic suppression should be explained better.

2. results are reported as aggregates; it would be great to have at least some sense of the across-subjects variability. How robust is the effect at the individual subject level?

3. because the intrasaccadic perception phenomenon is so easily explained by the pattern of retinal stimulation during saccadic eye rotation, I have no problems with an analysis approach that essentially disregards subjective reports and simply groups trials based on eye velocity. however, it would be interesting to know whether any relationship can be found between subjective ratings and pupil modulation. For example, did subjects who reported “seeing something” with highest confidence also have larger pupil modulations? this would help supporting the idea that pupil size provides information on subjective percepts (and can therefore be a valid support in many psychophysical studies!)




very minor
- please specify what you consider to be an eye-movement artefact vs. an actual eye-position dependent modulation of pupil size (p.13). pupillometry is seldom combined with eye movements, especially in 2D; even though this is clearly not the point of the study, the reported pupil traces are a valuable piece of data in itself and it is worth commenting on their peculiar shapes (e.g. vertical vs. horizontal)

- I really don’t want to impose this upon the authors, so please feel free to ignore this last comment. I just think that a reference to my TICS mini-review (Binda Murray TiCS 2015) would be appropriate; there, I specifically make the point that pupillometry has the potential to become a useful tool to investigate visual processing - not just retinal/subcortical disfunction as in the bed-side pupil test, but even the most sophisticated aspects of perception, of which saccadic suppression (and lack of) is an excellent example.

Best
Paola (Binda)

·

Basic reporting

The manuscript adheres to the PeerJ policies, is written in clear and comprehensive English, conforms to the template structure of PeerJ articles and provides informative figures.

However, I feel that the authors need to refine the introduction and discussion with regard to the motivation of the study and the conclusions made based on the results, respectively. Please see my full review in the "General Comments for the the Author".

Experimental design

The manuscript describes an original primary research and the investigations have been conducted rigorously and at high technical standard.

Validity of the findings

There are some minor statistical aspects that need revision. Additionally, some of the conclusions are not well motivated by the results of the study. Please see my full review in the "General Comments for the Author".

Comments for the author

In this manuscript, Mathôt et al. investigate intrasaccadic perception using pupil size as an objective measure. To this end, the authors presented a sinusoidal grating that changed polarity at a frequency larger than the flicker fusion threshold. Hence, the gratings were perceived as homogeneous surfaces when subjects fixated or performed saccades along the grating’s orientation. However, when subjects performed saccades perpendicular to the grating’s orientation, the retinal speed of the grating was reduced (canceled out) and an intrasaccadic percept became observable. The authors demonstrate that the pupil is sensitive to such intrasaccadic perception by showing that the pupil constricts more strongly for saccades in the “motion” direction of the grating than for saccades parallel to the grating’s orientation. The authors demonstrate that this effect of the intrasaccadic percept on pupil constrictions can be observed for horizontal and vertical saccades.
This is an interesting and well-conducted study, which would fit nicely to PeerJ. However, I believe that the following suggestions (listed below) will help the authors to improve the qaulity of the paper and make the manuscript suitable for publication in PeerJ.

MAJOR ISSUES
Introduction: I am not too satisfied with the introduction as the purpose of the study is not very well motivated. First, the authors extensively summarize the active-suppression and passive-masking hypotheses, yet, these do not play a major role in the paper. This part could be streamlined without considerable loss of information. Second, the authors describe the predictive coding hypothesis, but do not illustrate how their experiment would provide evidence confirming or refuting this hypothesis (see also my last “major issue”). Third (and most severely), the authors do not motivate the main aspect that differentiates their study from former studies, i.e., the use of pupil size as an objective measure of intrasaccadic perception. In my opinion, this does represent a solid contribution - but the authors so far do not sufficiently illustrate why it is important to overcome subjective report.
p. 13: The term “reliable” could be misleading as this might be confused with reliability in the sense of test-retest reliability (inter-subject stability of an effect across multiple experimental sessions). Why did the authors use that terminology? To prevent any misinterpretation, the authors should consider re-phrasing that term throughout the manuscript.
p. 14: Is the pupil size time course averaged across all trials of all subjects (fixed-effects)? It would be quite informative to see the mean pupil time courses for each subject (e.g., in the supplementary material). This would allow the reader to get a feeling of the between-subject variability of the effect.
p. 18: I wonder how the authors can conclude that “something odd” is equal to the “occurrence of intrasaccadic perception”. Their argument would be more convincing when illustrating a relation between the subjective rating and the effect of intrasaccadic perception on pupil constriction.
Discussion: Some of the claims made by the authors are rather speculative. In my opinion, the present results do not provide direct/hard evidence for the predictive coding theory or the fact that any postsaccadic pupil constriction is due to changes in the visual input. My feeling is that the authors need to highlight more convincingly how their findings directly speak to the above-mentioned conclusions (see also my first “major issue”).

MINOR COMMENTS
p. 4, l. 24: “…passive-masking hypothesis…“
p. 5, l. 18: “…why we do not see movements…”
p. 13: The argumentation for the author’s proceeding with regard to the position artifacts in pupil size is not clear enough.
p. 16: In my opinion, a paired t-test is not the correct way to test whether variances in peak velocity are different for vertical and horizontal saccades. Using a t-test, one can only assess whether two samples have the same mean. Therefore, the Mauchly-Test is what the authors should use.
p. 18: Please provide the respective test statistics when comparing the subjective ratings (as the authors speak about an “unreliable” effect).

---

## Round 0.2 · accepted · Accept

Dear Authors, Congratulations!

Your manuscript has been accepted by our peer reviewers thus it will now undergo processing prior to publication. Thank you for submitting your revised manuscript to us.

·

Basic reporting

No Comments

Experimental design

No Comments

Validity of the findings

No Comments

Comments for the author

The authors have addressed all the issues I raised and I have no further suggestions.

·

Basic reporting

The manuscript adheres to the PeerJ standards.

Experimental design

The manuscript adheres to the PeerJ standards.

Validity of the findings

The manuscript adheres to the PeerJ standards.

Comments for the author

I'm statisfied with the author's revision and recommed to accept the paper as it is.